# Polyoxypregnane Glycosides from Root of *Marsdenia tenacissima* and Inhibited Nitric Oxide Levels in LPS Stimulated RAW 264.7 Cells

**DOI:** 10.3390/molecules28020886

**Published:** 2023-01-16

**Authors:** Zhi Na, Pianchou Gongpan, Qingfei Fan

**Affiliations:** 1Key Laboratory of Tropical Plant Resources and Sustainable Use, Xishuangbanna Tropical Botanical Garden, Chinese Academy of Sciences, Mengla 666303, China; 2College of Science, Yunnan Agricultural University, Kunming 650201, China

**Keywords:** *Marsdenia tenacissima*, *Asclepiadaceae*, polyoxypregnane glycoside, anti-inflammatory

## Abstract

Six new polyoxypregnane glycosides, marstenacisside F1–F3 (**1**–**3**), G1–G2 (**4**–**5**) and H1 (**6**), as well as 3-*O*-β-D-glucopyranosyl-(1→4)-6-deoxy-3-*O*-methyl-β-D-allopyranosyl-(1→4)-β-D-oleandropyranosyl-11α,12β-di-*O*-benzoyl-tenacigenin B (**7**), were isolated from the roots of *Marsdenia tenacissima*. Their structures were established by an extensive interpretation of their 1D and 2D NMR and HRESIMS data. Compounds **1**–**7** were tenacigenin B derivatives with an oligosaccharide chain at C-3. This was the first time that compound **7** had been isolated from the title plant and its ^1^H and ^13^C NMR data were reported. Compounds **4** and **5** were the first examples of C_21_ steroid glycoside bearing unique β-glucopyranosyl-(1→4)-β-glucopyranose sugar moiety. All the isolated compounds were evaluated for anti-inflammatory activity by inhibiting nitric oxide (NO) production in the lipopolysaccharide-induced RAW 264.7 cells. The results showed that marstenacisside F1 and F2 exhibited significant NO inhibitory activity with an inhibition rate of 48.19 ± 4.14% and 70.33 ± 5.39%, respectively, at 40 μM, approximately equal to the positive control (L-NMMA, 68.03 ± 0.72%).

## 1. Introduction

*Marsdenia tenacissima* (Roxb.) Moon (*Asclepiadaceae*), a perennial climber, is distributed mainly in the southwest of China and other parts of tropical and subtropical Asia. The stems and roots of *M. tenacissima* are traditional Chinese medicine and Dai herbal medicine, respectively. The dried stems of *M. tenacissima*, known as “tongguanteng”, have been used in the treatment of asthma, cancer, and trachitis [1]. The roots of this plant, known as “Dai-Bai-Jie”, have been widely used as a Dai herbal medicine by Dai people living in Laos, Myanmar and the Yunnan province of China due to the root’s pharmacological functions of relieving pain, clearing heat, decreasing swelling and detoxification, etc. [2]. There have been many more chemical investigations on the stems than the roots of *M. tenacissima*. Previous phytochemical studies on the stems had revealed this plant as an extremely rich source of C_21_ steroid glycosides [3,4,5,6,7,8,9,10,11,12,13,14]. Although “Dai-Bai-Jie” has been widely used as a Dai herbal medicine, few phytochemical studies on the roots of this plant have been reported so far [15,16,17]. These studies showed that the main chemical composition of roots was also the same as the stems, i.e., C_21_ steroid glycosides. These compounds were only screened for anti-HIV activity and were necessary for anti-inflammatory activity, because the traditional usage for “Dai-Bai-Jie” was the treatment of inflammatory-associated diseases. Inflammation is a response of the organism to injury related to physical or chemical noxious stimuli or microbiological toxins, which is involved in multiple pathologies such as arthritis, asthma, multiple sclerosis, colitis, inflammatory bowel diseases, and atherosclerosis [18]. It can be speculated that the presence of key chemical constituents with effective anti-inflammatory activity had led to the extensive clinical application of “Dai-Bai-Jie” in traditional ethnomedicine, so we were interested in clarifying the relationship between the constituents and anti-inflammatory activity of this plant.

In order to search for more novel natural products, particularly those with potential anti-inflammatory activity, from the roots of *M. tenacissima*, a systematic phytochemical study was carried out on their 95% ethanol extract. As a result, six new polyoxypregnane glycosides, named marstenacisside F1–F3 (**1**–**3**), G1–G2 (**4**–**5**), and H1 (**6**), as well as 3-*O*-β-D-glucopyranosyl-(1→4)-6-deoxy-3-*O*-methyl-β-D-allopyranosyl-(1→4)-β-D-oleandropyranosyl-11α,12β-di-*O*-benzoyl-tenacigenin B (**7**), were isolated (Figure 1). Compound **7** was isolated from the title plant, and its ^1^H and ^13^C NMR data were reported, for the first time. In the present paper, we describe the isolation and structure elucidation of these compounds, and we also evaluate the anti-inflammatory activity of the isolated compounds in terms of the inhibitory effect on NO production in LPS-induced RAW 264.7 cells.

## 2. Results and Discussion

### 2.1. Structural Elucidation

Compound **1** displayed a sodium adduct ion at *m/z* 877.4344 [M + Na]^+^ in its HRESIMS spectrum (Appendix A), and its molecular formula was determined as C_47_H_66_O_14_ (calcd for C_47_H_66_NaO_14_, 877.4345). There were three methyl signals at δ_H_ 1.18 (3H, s, CH_3_-18), 1.09 (3H, s, CH_3_-19), and 2.26 (3H, s, CH_3_-21), and three methine protons bearing secondary alcoholic functions at δ_H_ 3.60 (m, H-3), 5.58 (t, *J* = 10.2 Hz, H-11) and 5.25 (d, *J* = 10.2 Hz, H-12) in the ^1^H NMR spectrum of **1**. The combination of the ^1^H and ^13^C NMR data (Appendix A) indicated a C_21_ steroidal skeleton for **1**. By comparison with C_21_ steroids isolated from the title plant, the ^13^C NMR data (Table 1) of **1** were similar to those of 11,12-diester of tenacigenin B [19]. The signals at δ_H_ 6.55 (q, *J* = 6.8 Hz), 1.46 (m), and 1.44 (s), and δ_C_ 167.4, 138.1, 128.5, 14.2, and 11.6, indicated the existence of a tigloyl (Tig) group. The long-range correlation from δ_H_ 5.58 (H-11) to δ_C_ 167.4 (Tig-C-1) in the HMBC spectrum (Figure 2) was observed, which indicated the Tig group was assigned at C-11. At the same time, there was a benzoyl (Bz) group in **1**, due to the existence of a series of NMR signals at δ_H_ 7.89 (d, *J* = 7.4 Hz), 7.51 (t, *J* = 7.4 Hz), and 7.38 (t, *J* = 7.4 Hz), and δ_C_ 166.1, 133.1, 129.7 (2C), 129.4 and 128.3 (2C). The Bz group was demonstrated to be attached at C-12 by the HMBC correlation of H-12 (δ_H_ 5.25) to the carbonyl carbon (δ_C_ 166.1) of the benzoyl group. In the NOESY spectrum of **1** (Figure 3), crossing peaks between H-11 and CH_3_-18 (δ_H_ 1.18) and between H-12 and H-9 (δ_H_ 2.08) revealed that H-11 and H-12 were in β-orientation and α-orientation, respectively. Furthermore, the C-17 side chain was in an α-orientation as supported by the NOESY correlations between H-17 (δ_H_ 2.95) and CH_3_-18 (δ_H_ 1.18), and between H-12 (δ_H_ 5.25) and CH_3_-21 (δ_H_ 2.26) (Appendix A). Thus, the aglycone of **1** was determined to be 11α-*O*-tigloyl-12β-*O*-benzoyl-tenacigenin B.

In the NMR spectra of **1**, there were two anomeric proton signals at δ_H_ 4.58 (dd, *J* = 9.8, 1.8 Hz) and 4.79 (d, *J* = 8.1 Hz) and two carbon signals at δc 96.9 and 99.1. The above evidence proved that the sugar moiety of **1** contained two units. The coupling constants (8.1 and 9.8 Hz) of the two anomeric protons indicated that both glycosidic linkages were β-oriented. At the same time, there were characteristic proton signals of two methyls at δ_H_ 1.36 (d, *J* = 5.5 Hz) and 1.25 (d, *J* = 6.0 Hz); two methoxyl groups at δ_H_ 3.66 (s) and 3.37 (s) in the ^1^H NMR spectrum of **1**. The evidence of two methoxyl groups located at C-3 in each of the two sugar moiety was deduced from the long-range correlations between the methoxyl group at δ_H_ 3.37 (s) and the carbon signal at δ_C_ 78.8, and between another methoxyl group at δ_H_ 3.66 (s) and the carbon signal at δ_C_ 81.0. Subsequently, the two sugar units were identified as 6-deoxy-3-methoxy sugars, which generally occur in *M. tenacissima* [1]. Based on the HSQC, HMBC, and ^1^H-^1^H COSY spectra (Appendix A), the NMR spectra of each sugar were fully assigned (Table 2 and Table 3). The two sugar units were then determined as oleandropyranosyl (Ole) and 6-deoxy-3-*O*-methyl-allopyranose (Allo), respectively [20]. Meanwhile, the oleandrose was the inner sugar and 6-deoxy-3-*O*-methyl allose was the outer, which was supported by the HMBC correlations of δ_H_ 4.79 (Allo-H-1) with δ_C_ 79.1 (Ole-C-4), and δ_H_ 4.58 (Ole-H-1) with δ_C_ 76.4 (aglycone-C-3). Consequently, the sugar moiety of **1** was determined as pachybiose. Compared to the previously reported data of tenacigenin B [21], changes in the chemical shift in aglycone of **1**, i.e., C-2 (−3.2 ppm), C-3 (+5.6 ppm), and C-4 (−3.8 ppm) were observed, which suggested that the sugar moiety was linked to the C-3 hydroxyl of the aglycone. Thus, compound **1** was elucidated as 3-*O*-6-deoxy-3-*O*-methyl-β-D-allopyranosyl (1→4)-β-D-oleandropyranosyl-11α-*O*-tigloyl-12β-*O*-benzoyl-tenacigenin B and named marstenacisside F1.

The ^1^H and ^13^C NMR spectroscopic data ascribed to the sugar moieties of **2**–**3** are consistent with those of **1** (Table 2 and Table 3), so they should contain the same sugar moiety as **1**.

Compound **2** possessed a molecular formula of C_49_H_64_O_14_, determined by HRESIMS ion at *m/z* 899.4190 [M + Na]^+^ (calcd for C_49_H_64_NaO_14_, 899.4188) (Appendix A). The NMR data of **2** showed a pattern analogous to **1,** except for an ester group (Appendix A). In the ^13^C NMR spectrum of **2**, there were signals of two benzoyl groups on the aglycone of **2**. Meanwhile, the tigloyl unit signals of **1** were absent in **2**. An extra Bz group was positioned at C-11, which was deduced from the HMBC correlation from δ_H_ 5.77 (H-11) to δ_C_ 166.2 (Bz_1_-C-1) (Appendix A). Due to the HMBC correlation between δ_H_ 4.58 (Ole-H-1) and δ_C_ 76.4 (aglycone-C-3), the glycosidation site was located at C-3 of the aglycone. Consequently, **2** was defined as 3-*O*-6-deoxy-3-*O*-methyl-β-D-allopyanosyl(1→4)-β-D-oleandropyranosyl-11α,12β-di-*O*-benzoyl-tenacigenin B, and named marstenacisside F2 (Appendix A).

Compound **3** showed a quasi-molecular ion peak at *m/z* 825.4040 [M + Na]^+^, in accordance with the molecular formula C_43_H_62_O_14_ (calcd for C_43_H_62_NaO_14_, 825.4032) (Appendix A). In the ^1^H NMR spectrum of **3** (Appendix A), due to the absence of an ester group at C-12, there was a higher-field shift signal at δ_H_ 3.27 (1H, d, *J* = 9.7 Hz), compared to the δ_H_ 5.35 (1H, d, *J* = 10.2 Hz) in the ^1^H NMR spectrum of **2**. Consequently, the aglycone of **3** was a monoester of tenacigenin B. The signals at δ_H_ 7.09 and 6.75 (d, *J* = 8.6 Hz, each 2H) and the ^13^C NMR (Appendix A) singlet at δ_C_ 155.2 indicated that **3** contained a 4-hydroxyphenyl group. In the HMBC spectrum (Appendix A), the aromatic resonances at δ_C_ 125.3 and 130.7 were correlated with the signal at δ_H_ 3.54 (m, 2H), and 7.09 correlated with the δ_C_ 41.1. Moreover, the HMBC spectrum showed a correlation between δ_H_ 3.54 and δ_C_ 172.0. Therefore, a methylene group (δ_H_ 3.54, δ_C_ 41.1) was located between the 4-hydroxyphenyl moiety and the carbonyl group (δ_C_ 172.0). Hence, **3** contained a (4-hydroxyphenyl) acetyl (HPA) group. The HPA group was assigned at C-11 by HMBC correlation of the proton at δ_H_ 5.05 (H-11) with the carbonyl carbon at δ_C_ 172.0 of the HPA group. The glycosidation site was deduced from the HMBC cross peaks between δ_H_ 4.58 (Ole-H-1) and δ_C_ 76.6 (aglycone-C-3). Therefore, **3** was defined as 3-*O*-6-deoxy-3-*O*-methyl-β-D-allopyanosyl (1→4)-β-D-oleandropyranosyl-11α-*O*-(4-hydroxyphenyl) acetyl-tenacigenin B and named marstenacisside F3 (Appendix A).

Compound **4** had a molecular formula of C_40_H_60_O_17,_ as determined by HRESIMS [M + Na]^+^ ion at *m/z* 835.3714 (calcd for C_40_H_60_NaO_17_, 835.3723) (Appendix A). The proton signals of δ_H_ 1.90 (3H, s), 6.86 (1H, qq, *J* = 7.1, 1.4 Hz), 1.80 (3H, s), and 1.61 (3H, d, *J* = 7.1 Hz) and the carbon signals of δ_C_ 170.7, 20.5, 167.2, 129.2, 138.1, 12.2, and 14.3 indicated the presence of an acetyl (Ac) and a tigloyl group in the agcylone of **4**. The ^1^H NMR data of the aglycone moiety of **4** were very close to those of **1**, except for the presence of the signal for an acetyl group at δ_H_ 1.90 (3H, s) and the absence of the protons signals for the benzoyl group. HMBC cross-peaks between δ_H_ 5.76 (H-11) and δ_C_ 167.2 (Tig-C-1), and between δ_H_ 5.40 (H-12) and δc 170.7 (Ac-C-1) (Appendix A), indicated that a tigloyl and acetyl group were located at C-11 and C-12, respectively. Accordingly, the aglycone of **4** was identified as 11α-*O*-tigloyl-12β-*O*-acetyl-tenacigenin B.

In the NMR spectra of **4** (Appendix A), there were two anomeric proton signals at δ_H_ 5.01 (d, *J* = 7.6 Hz) and 5.22 (d, *J* = 7.8 Hz) and two carbon signals at δc 101.2 and 106.7. The above facts proved that the sugar moiety of **4** contained two units. Furthermore, the ^13^C NMR spectra displayed two terminal oxygenated methylene groups at δc 62.6. On the basis of the above evidence and compared with previously reported data [22], the two sugar units were identified as glucopyranoses. According to the coupling constants of anomeric protons (7.6 and 7.8 Hz), the linkages of two sugar units were in β-configuration. The linkage of two sugar moiety could be β-glucopyranosyl-(1→4)-β-glucopyranoside deduced from the long-range correlations between δ_H_ 5.22 (Glc_2_-H-1) and δc 84.7 (Glc_1_-C-4). Relative to the previously reported values of 11α-O-tigloyl-12β-O-acetyl-tenacigenin B [23], changes in the chemical shift in aglycone of **4**, i.e., C-2 (−1.8 ppm), C-3 (+8.4 ppm), and C-4 (−2.9 ppm) were detected, which suggested that the sugar moiety was attached at the C-3 of the aglycone. Thus, compound **4** was finally elucidated as 3-*O*-β-D-glucopyranosyl-(1→4)-β-D-glucopyranosyl-11α-*O*-tigloyl-12β-*O*-acetyl-tenacigenin B and named marstenacisside G1 (Appendix A).

The ^1^H and ^13^CNMR data of the sugar moiety of **5** were well in agreement with those of **4**. Accordingly, **5** had the same sugar moiety as **4**.

Compound **5** exhibited a molecular formula of C_40_H_62_O_17_ based on the HRESIMS ion [M + Na]^+^ at *m/z* 837.3883 (calcd for C_40_H_62_NaO_17_, 837.3879) (Appendix A). The NMR data of **5** (Appendix A) showed a similar pattern to **4**, except for an ester substitution. In the ^1^H NMR spectrum of **5**, there were proton signals of an acetyl group at δ_H_ 2.02 (3H, s) and extra signals of 2-methylbutyryl ester units at δ_H_ 0.84 (3H, t, *J* = 7.5 Hz), 1.04 (3H, d, *J* = 7.0 Hz), 1.34 (1H, m), 1.68 (1H, m), and 2.26 (1H, m). Meanwhile, the signals of the tigloyl unit of **4** were absent in **5**. The HMBC cross peak between the carbonyl carbon at δ_C_ 175.5 of the 2-methylbutyryl group and the proton signal at δ_H_ 5.68 (H-11) disclosed a 2-methylbutyryl group located at C-11 (Appendix A). Likewise, the HMBC correlation of the carbonyl carbon at δ_C_ 170.8 of the acetyl unit with the proton signal at δ_H_ 5.36 (H-12) revealed the existence of an acetyl group at C-12. The glycosidation site was deduced from the long-range coupling of the δ_H_ 5.08 (Glc_1_-H-1) with δ_C_ 77.9 (agclone-C-3). Accordingly, compound **5** was established as 3-*O*-β-D-glucopyranosyl-(1→4)-β-D-glucopyranosyl-11α-*O*-2-methylbutyryl-12β-*O*-acetyl-tenacigenin B and was named marstenacisside G2 (Appendix A).

Compound **6** gave a molecular formula of C_51_H_74_O_20_ based on the HRESIMS (*m/z* 1029.4659 [M + Na]^+^, calcd for C_51_H_74_NaO_20_, 1029.4666) (Appendix A). ^13^C NMR data analysis indicated that the aglycone moiety of **6** differs from the aglycone moiety of **3** by the presence of an extra acetyl group (δ_C_ 170.8 and 20.3). The positions of the diester groups were deduced by the HMBC correlations between δ_H_ 5.68 (H-11) and δ_C_ 171.5 (HPA-C-1), and between δ_H_ 5.36 (H-12) and δ_C_ 170.8 (Ac-C-1) (Appendix A). Therefore, the aglycone structure of **6** was identified as 11α-*O*-(4-hydroxyphenyl) acetyl-12β-*O*-acetyl-tenacigenin B.

The ^1^H and ^13^C NMR spectra of **6** (Appendix A) exhibited three anomeric proton signals at δ_H_ 5.27 (1H, d, *J* = 8.1 Hz), 4.97 (1H, d, *J* = 7.8 Hz), and 4.78 (1H, d, *J* = 8.9 Hz) and three anomeric carbon signals at δ_C_ 102.0, 106.6 and 97.5, suggesting the existence of three sugar units in the molecule. Due to the large coupling constants of anomeric protons, the linkages of three sugar units were in β-configuration. At the same time, there were characteristic proton signals of two methyls at δ_H_ 1.67 (d, *J* = 5.8 Hz) and 1.63 (d, *J* = 6.2 Hz); two methoxyl groups at δ_H_ 3.81 (s) and 3.50 (s); and two ABM spin protons at δ_H_ 4.36 (dd, *J* = 11.6, 5.4 Hz), 4.54 (dd, *J* = 11.6, 2.3 Hz) in the ^1^H NMR spectrum of **6**. On the basis of the above evidence and compared with previously reported data [24,25], the sugar units were identified as oleandrose, 6-deoxy-3-*O*-methyl-allose, and glucoses. The connectivity of the sugars was established by the HMBC correlations between δ_H_ 4.97 (Glc-H-1) and δ_C_ 83.4 (Allo-C-4); between δ_H_ 5.27 (Allo-H-1) and δ_C_ 83.4 (Ole-C-4). As a result, the sugar moiety was determined as β-D-glucopyranosyl-(1→4)-6-deoxy-3-*O*-methyl-β-D-allopyranosyl-(1→4)-β-D-oleandropyranoside, which was identical to the neocondurangotriose in the compounds that were also isolated from *M. tenacissima* [17]. Furthermore, the glycosidation site was deduced from the long-range correlations between δ_H_ 4.78 (Ole-H-1) and δ_C_ 76.0 (aglycone-C-3). Hence, the structure of **6** was established as 3-*O*-β-D-glucopyranosyl-(1→4)-6-deoxy-3-*O*-methyl-β-D-allopyranosyl-(1→4)-β-D-oleandropyranosyl-11α-*O*-(4-hydroxyphenyl) acetyl-12β-*O*-acetyl-tenacigenin B and named marstenacisside H1 (Appendix A).

Compound **7** had a molecular formula of C_55_H_74_O_19_ determined by HRESIMS ion [M + Na]^+^ at *m*/*z* 1061.4718 (calcd for C_55_H_74_NaO_19_, 1061.4722) (Appendix A). Compound **7** was predicted to be novel pregnane glycoside in a crude extract of *Marsdenia tenacissima* by means of LC-ESI-MS^n^ [26], and the compound was not isolated from the crude extract. This is the first time that compound **7** has been isolated from the title plant and its ^1^H and ^13^C NMR data were reported. The structure of **7** was eluciated as 3-*O*-β-D-glucopyranosyl-(1→4)-6-deoxy-3-*O*-methyl-β-D-allopyranosyl-(1→4)-β-D-oleandropyranosyl-11α,12β-di-*O*-benzoyl-tenacigenin B, on the basis of 1D, 2D NMR and HRESI data (Appendix A).

### 2.2. NO Inhibitory Evaluations

Compounds **1**–**7** were screened for anti-inflammatory activity by inhibiting NO production in LPS-induced RAW 264.7 cells. Compounds **1** and **2** showed significant NO inhibitory activity with an inhibition rate of 48.19 ± 4.14% and 70.33 ± 5.39%, respectively, at 40 μM, approximately equal to the positive control (L-NMMA, 68.03 ± 0.72%) (Figure 4 and Table 4). The effects of compounds **1**–**7** on cell viability are shown in Appendix A. All compounds showed dose-dependent NO inhibitory activity. Only **1** and **2** showed significant NO inhibitory activity, and other compounds did not show the activity. The above facts may be related to the structure of compounds **1** and **2**. There were tigloyl and/or benzoyl groups at C-11 and C-12 of **1** and **2**, and the sugar moiety was pachybiose. Although **7** also had two benzoyl groups at C-11 and C-12, the sugar moiety was neocondurangotriose.

## 3. Materials and Methods

### 3.1. General Experimental Procedures

The UV spectra were collected on a Shimadzu UV-2401 PC spectrophotometer (Shimadzu Corp: Kyoto, Japan). Optical rotations were measured on an Autopol VI polarimeter; The IR spectra were determined on a Nicolet iS10 spectrometer with KBr pellets. HRESIMS was recorded on an Agilent 1290 UPLC/6540 Q-TOF mass spectrometer (Agilent, Palo Alto, CA, USA). All NMR spectra were acquired on a Bruker Avance III 500 spectrometer. Semi-preparative HPLC was performed on a Waters HPLC system consisting of a 1525 binary pump and a 2487 detector, equipped with a YMC-pack ODS-A column (250 × 10 mm, YMC Co., Ltd., Kyoto, Japan). Silica gel (200–300 mesh, Qingdao Marine Chemical Co., Ltd., Qingdao, China), Lichroprep RP-18 gel (40–63 μm, Merck, Darmstadt, Germany), Sephadex LH-20 gel (GE Healthcare, Sweden), and MCI gel (75–150 μm, Mitsubishi Chemical Co., Tokyo, Japan) were used to perform column chromatography.

### 3.2. Plant Material

The roots of *Marsdenia tenacissima* were collected from Simao, Yunnan Province, China in January 2020. A voucher specimen (No. 20200101) was deposited in the authors’ research group.

### 3.3. Extraction and Isolation

The dried powder roots of *Marsdenia tenacissima* (2.5 kg) were percolated with 95% ethanol at room temperature three times (3 days each time) and then concentrated under reduced pressure to give concentrated extract. The concentrated extract was efficiently partitioned with ethyl acetate (EtOAc). The EtOAc fraction (68.6 g) was separated by an MCI gel CHP 20P column, eluted with MeOH–H_2_O (*v*/*v*, 30:70, 50:50, 80:20, 95:5) to provide four portions (Fr. A–D). Fr. B (28.1 g) was subjected to silica gel CC eluted with CH_2_Cl_2_–MeOH (25:1–3:1) to obtain five fractions (Fr. B.1–5). Fr. B.2 (2.8 g) was chromatographed over a Sephadex LH-20 column, eluting with MeOH to give four fractions (Fr. B.2.1–4). Fr. B.2.2 (256 mg) was further purified by semi-preparative HPLC using MeOH/H_2_O (70:30, 3 mL/min) to give compounds **3** (12 mg, *t*_R_ = 12.3 min) and **6** (9 mg, *t*_R_ = 19.6 min). Fr. B.4 (2.1 g) was separated by ODS MPLC (MeOH–H_2_O, 60:40 to 100:0) to yield five fractions (Fr. B.4.1–5). Fr. B.4.3 (202 mg) was further separated by semi-preparative with MeOH/H_2_O (80:20, 3 mL/min) to yield compounds **4** (11 mg, *t*_R_ = 8.3 min) and **5** (8 mg. *t*_R_ = 9.6 min). Fr. C (35.7 g) was subjected to silica gel CC eluted with CH_2_Cl_2_–MeOH (50:1–4:1) to obtain five fractions (Fr. C.1–5). Fr. C.3 (3.2 g) was chromatographed over a Sephadex LH-20 column, eluting with CH_2_Cl_2_–MeOH (1:1) to give four fractions (Fr. C.3.1–4). Fr. C.3.3 (306 mg) was further purified by semi-preparative HPLC using MeOH/H_2_O (75:25, 3 mL/min) to afford compounds **7** (11 mg, *t*_R_ = 12.3 min), **1** (9 mg, *t*_R_ = 16.6 min), and **2** (10 mg, *t*_R_ = 20.7 min).

### 3.4. Compound Characterization Data

Marstenacisside F1 (**1**): white amorphous powder; [α${]}_{D}^{23}$ + 19.6 (*c* 0.15, MeOH); UV (MeOH) λ_max_ (log ε): 196 (4.46), 226 (4.26), 273 (3.01) nm; IR (KBr): υ_max_ 3436, 2929, 1719, 1451, 1367, 1281, 1164, 1071, 988, 711 cm^−1^; ^1^H NMR (CDCl_3_) data of aglycone moiety of **1**: δ 1.09 (3H, s, 19-CH_3_), 1.18 (3H, s, 18-CH_3_), 1.44 (3H, brs, Tig-H-5), 1.46 (3H, m, Tig-H-4), 2.08 (1H, m, H-9), 2.26 (3H, s, 21-CH_3_), 2.99 (1H, d, *J* = 6.4 Hz, H-17β), 3.60 (1H, m, H-3), 5.25 (1H, d, *J* = 10.2 Hz, H-12α), 5.58 (1H, t, *J* = 10.2 Hz, H-11β), 6.55 (1H, q, *J* = 6.8 Hz, Tig-H-3), 7.38 (2H, t, *J* = 7.4 Hz, Bz-H-4, 6), 7.51 (1H, t, *J* = 7.4 Hz, Bz-H-5), 7.89 (2H, d, *J* = 7.4 Hz, Bz-H-3, 7); HRESIMS: *m/z* 877.4344 [M + Na]^+^ (calcd for C_47_H_66_NaO_14_, 877.4345); for ^13^C NMR data of the aglycone moiety of **1** see Table 1. For ^1^H and ^13^C NMR data of the sugar moiety of **1** see Table 2 and Table 3.

Marstenacisside F2 (**2**): white amorphous powder; [α${]}_{D}^{23}$ + 26.9 (*c* 0.18, MeOH); UV (MeOH) λ_max_ (log ε): 196 (4.49), 230 (4.33), 274 (3.34) nm; IR (KBr): υ_max_ 3436, 2932, 1721, 1451, 1367, 1283, 1162, 1070, 988, 708 cm^−1^; ^1^H NMR (CDCl_3_) data of aglycone moiety of **2**: δ 1.14 (3H, s, 19-CH_3_), 1.21 (3H, s, 18-CH_3_), 2.21 (1H, m, H-9), 2.27 (3H, s, 21-CH_3_), 3.01 (1H, d, *J* = 7.3 Hz, H-17β), 3.60 (1H, m, H-3), 5.35 (1H, d, *J* = 10.2 Hz, H-12α), 5.77 (1H, t, *J* = 10.2 Hz, H-11β), 7.18 (2H, t, *J* = 7.4 Hz, Bz_2_-H-4, 6), 7.22 (2H, t, *J* = 7.4 Hz, Bz_1_-H-4, 6), 7.33 (1H, t, *J* = 7.4 Hz, Bz_2_-H-5), 7.37 (1H, t, *J* = 7.4 Hz, Bz_1_-H-5), 7.75 (4H, d, *J* = 7.4 Hz, Bz_1_-H-3, 7, Bz_2_-3, 7); HRESIMS: *m/z* 899.4190 [M + Na]^+^ (calcd for C_49_H_64_NaO_14_, 899.4188); for ^13^C NMR data of the aglycone moiety of **2** see Table 1. For ^1^H and ^13^C NMR data of the sugar moiety of **2** see Table 2 and Table 3.

Marstenacisside F3 (**3**): white amorphous powder; [α${]}_{D}^{19}$ − 4.9 (*c* 0.10, MeOH); UV (MeOH) λ_max_ (log ε): 196 (4.16), 224 (3.69), 278 (3.08) nm; IR (KBr): υ_max_ 3445, 2933, 1703, 1619, 1367, 1164, 1068, 989, 610 cm^−1^; ^1^H NMR (CDCl_3_) data of aglycone moiety of **3**: δ 1.03 (3H, s, 19-CH_3_), 1.14 (3H, s, 18-CH_3_), 1.85 (1H, m, H-9), 2.26 (3H, s, 21-CH_3_), 3.01 (1H, t, *J* = 6.2 Hz, H-17β), 3.27 (1H, d, *J* = 9.7 Hz, H-12α), 3.54 (2H, m, HPA-2), 3.66 (1H, m, H-3), 5.05 (1H, d, *J* = 9.7 Hz, H-11β), 6.75 (2H, d, *J* = 8.6 Hz, HPA-H-5, 7), 7.09 (2H, d, *J* = 8.6 Hz, HPA-H-4, 8); HRESIMS: *m/z* 825.4040 [M + Na]^+^ (calcd for C_43_H_62_NaO_14_, 825.4032); for ^13^C NMR data of the aglycone moiety of **3** see Table 1. For ^1^H and ^13^C NMR data of the sugar moiety of **3** see Table 2 and Table 3.

Marstenacisside G1 (**4**): white amorphous powder; [α${]}_{D}^{23}$ + 2.1 (*c* 0.10, MeOH); UV (MeOH) λ_max_ (log ε): 196 (3.93), 216 (3.94) nm; IR (KBr): υ_max_ 3428, 2934, 1706, 1619, 1368, 1269, 1077, 1031 cm^−1^; ^1^H NMR (pyridine-*d_5_*) data of aglycone moiety of **4**: δ 1.20 (3H, s, 19-CH_3_), 1.28 (3H, s, 18-CH_3_), 1.61 (3H, d, *J* = 7.1 Hz, Tig-H-5), 1.80 (3H, brs, Tig-H-4), 2.05 (1H, m, H-9), 1.90 (3H, s, Ac-H-2), 2.23 (3H, s, 21-CH_3_), 2.90 (1H, d, *J* = 6.7 Hz, H-17β), 3.88 (1H, m, H-3), 5.40 (1H, d, *J* = 10.2 Hz, H-12α), 5.76 (1H, t, *J* = 10.2 Hz, H-11β), 6.86 (1H, qq, *J* = 7.1, 1.4 Hz, Tig-H-3); HRESIMS: *m/z* 835.3714 [M + Na]^+^ (calcd for C_40_H_60_NaO_17_, 835.3723); for ^13^C NMR data of the aglycone moiety of **4** see Table 1. For ^1^H and ^13^C NMR data of the sugar moiety of **4** see Table 2 and Table 3.

Marstenacisside G2 (**5**): [α${]}_{D}^{22}$ + 13.3 (c 0.10, MeOH); UV (MeOH) λ_max_ (log ε): 196 (3.83), 228 (3.48), 274 (2.72) nm; IR (KBr): υ_max_ 3426, 2936, 1736, 1632, 1367, 1247, 1076, 1030 cm^−1^; ^1^H NMR (pyridine-*d_5_*) data of aglycone moiety of **5**: δ 0.84 (3H, t, *J* = 7.5 Hz, mBu-H-4), 1.04 (3H, d, *J* = 7.0 Hz, mBu-H-5), 1.13 (3H, s, 19-CH_3_), 1.25 (3H, s, 18-CH_3_), 1.34 and 1.68 (2H, m, mBu-H-3), 2.02 (3H, s, Ac-H-2), 2.00 (1H, m, H-9), 2.26 (1H, m, mBu-H-2), 2.24 (3H, s, 21-CH_3_), 2.87 (1H, d, *J* = 6.9 Hz, H-17β), 3.84 (1H, m, H-3), 5.36 (1H, d, *J* = 10.2 Hz, H-12α), 5.68 (1H, t, *J* = 10.2 Hz, H-11β); HRESIMS: *m/z* 837.3883 [M + Na]^+^ (calcd for C_40_H_62_NaO_17_, 837.3879); for ^13^C NMR data of the aglycone moiety of **5** see Table 1. For ^1^H and ^13^C NMR data of the sugar moiety of **5** see Table 2 and Table 3.

Marstenacisside H1 (**6**): white amorphous powder; [α${]}_{D}^{22}$ − 1.6 (*c* 0.10, MeOH); UV (MeOH) λ_max_ (log ε): 197 (4.38), 225 (3.98), 277 (3.57) nm; IR (KBr): υ_max_ 3445, 2935, 1738, 1516, 1367, 1253, 1071, 991 cm^−1^; ^1^H NMR (pyridine-*d_5_*) data of aglycone moiety of **6**: δ 1.13 (3H, s, 18-CH_3_), 1.17 (3H, s, 19-CH_3_), 1.77 (3H, s, Ac-H-2), 2.07 (1H, d, *J* = 10.1 Hz, H-9), 2.24 (3H, s, 21-CH_3_), 2.86 (1H, d, *J* = 6.7 Hz, H-17β), 3.82 (1H, m, H-3), 5.36 (1H, d, *J* = 10.2 Hz, H-12α), 5.68 (1H, t, *J* = 10.2 Hz, H-11β), 7.11 (2H, d, *J* = 8.4 Hz, HPA-H-5, 7), 7.29 (2H, d, *J* = 8.6 Hz, HPA-H-4, 8); HRESIMS: *m/z* 1029.4659 [M + Na]^+^ (calcd for C_51_H_74_NaO_20_, 1029.4666); for ^13^C NMR data of the aglycone moiety of **6** see Table 1. For ^1^H and ^13^C NMR data of the sugar moiety of **6** see Table 2 and Table 3.

3-*O*-β-D-glucopyranosyl-(1→4)-6-deoxy-3-*O*-methyl-β-D-allopyranosyl-(1→4)-β-D-oleandropyranosyl-11α,12β-di-*O*-benzoyl-tenacigenin B (**7**): white amorphous powder; [α${]}_{D}^{21}$ + 20.8 (*c* 0.23, MeOH); UV (MeOH) λ_max_ (log ε): 201 (4.18), 230 (4.24), 274 (3.16) nm; IR (KBr): υ_max_ 3427, 2934, 1721, 1452, 1367, 1281, 1071, 710 cm^−1^; ^1^H NMR (pyridine-*d_5_*) data of aglycone moiety of **7**: δ 1.29 (3H, s, 18-CH_3_), 1.31 (3H, s, 19-CH_3_), 2.09 (1H, m, H-9), 2.32 (3H, s, 21-CH_3_), 2.97 (1H, d, *J* = 6.3 Hz, H-17β), 3.78 (1H, m, H-3), 5.78 (1H, d, *J* = 10.2 Hz, H-12α), 6.13 (1H, t, *J* = 10.2 Hz, H-11β), 7.16 (2H, t, *J* = 7.4 Hz, Bz_2_-H-4, 6), 7.22 (2H, t, *J* = 7.4 Hz, Bz_1_-H-4, 6), 7.24 (1H, t, *J* = 7.4 Hz, Bz_2_-H-5), 7.43 (1H, t, *J* = 7.4 Hz, Bz_1_-H-5), 7.94 (2H, d, *J* = 7.4 Hz, Bz_2_-H-3, 7), 7.99 (2H, d, *J* = 7.4 Hz, Bz_1_-H-3, 7); HRESIMS: *m/z* 1061.4718 [M + Na]^+^ (calcd for C_55_H_74_NaO_19_, 1061.4722); for ^13^C NMR data of the aglycone moiety of **7** see Table 1. For ^1^H and ^13^C NMR data of the sugar moiety of **7** see Table 2 and Table 3.

### 3.5. Cell Culture and Nitric Oxide Inhibitory Assay

The macrophage RAW 264.7 cells (passage number was 10–13) were obtained from Cell Bank of Chinese Academy of Sciences. The RAW 264.7 cells were plated in 96-well plates (1.5 × 10^5^ cells/well) and treated with different isolate concentrations (dissolved in DMSO) of 10, 20, and 40 μM, respectively, followed by stimulation with 1 μg/mL LPS (Sigma, St. Louis, MO, USA) for 18 h [27]. Griess reagents (Sigma, St. Louis, MO, USA) were used to measure NO production. The optical density (OD) was determined at a 570 nm wavelength, with L-NMMA as a positive control [28]. Three independent experiments were carried out in triplicate. The cell viability was evaluated by the MTT assay [29].

## 4. Conclusions

Six new polyoxypregnane glycosides, marstenacisside F1–F3 (**1**–**3**), G1–G2 (**4**–**5**), and H1 (**6**), as well as 3-*O*-β-D-glucopyranosyl-(1→4)-6-deoxy-3-*O*-methyl-β-D-allopyranosyl-(1→4)-β-D-oleandropyranosyl-11α,12β-di-*O*-benzoyl-tenacigenin B (**7**), were isolated from the ethanolic extract of the roots of *Marsdenia tenacissima* by modern chromatographic techniques and characterized by comprehensive spectroscopic data. Their structures were tenacigenin B derivatives with an oligosaccharide chain at C-3. Compounds **4** and **5** were the first examples of C_21_ steroid glycoside bearing unique β-glucopyranosyl-(1→4)-β-glucopyranose sugar moiety. Compound **7** was isolated from the title plant for the first time, and its ^1^H and ^13^C NMR data were reported. The patterns of compounds **1**–**7** were consistent with those of compounds previously isolated from this plant. All isolates were evaluated for anti-inflammatory activity by inhibiting the production of NO stimulated by LPS in RAW 264.7 cells, with L-NMMA as a positive control. Among those compounds, compounds **1** and **2** exhibited significant NO inhibition at 40 μM.

## Figures and Tables

**Figure 1 molecules-28-00886-f001:**
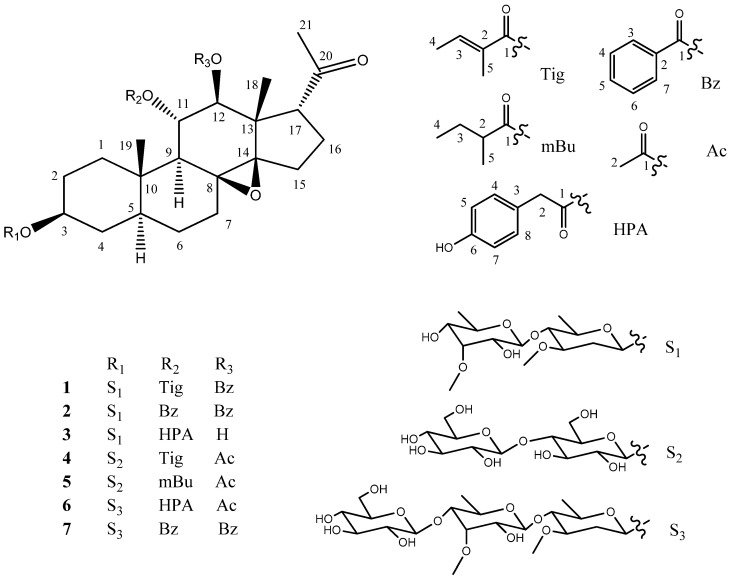
Structures of pregnane glycosides (**1**–**7**) isolated from *M. tenacissima*.

**Figure 2 molecules-28-00886-f002:**
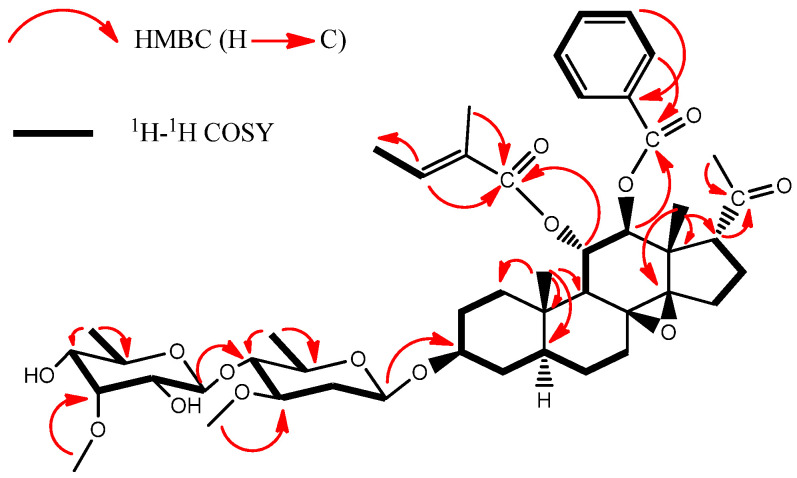
Key HMBC and 1H-1H COSY correlations of **1**.

**Figure 3 molecules-28-00886-f003:**
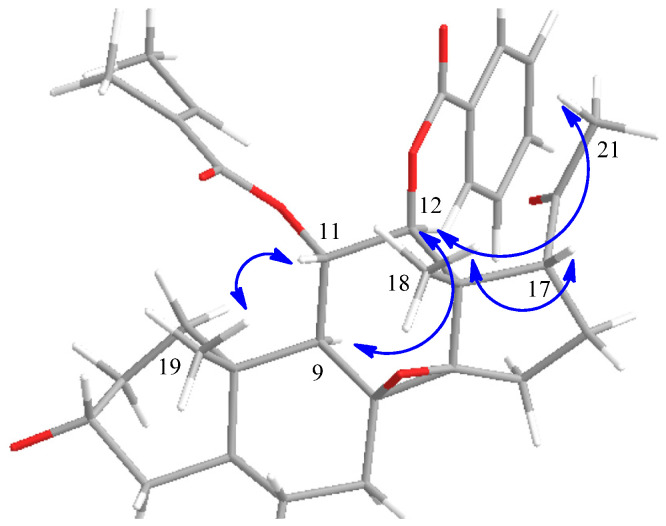
Key ROESY correlations of the aglycone of **1**.

**Figure 4 molecules-28-00886-f004:**
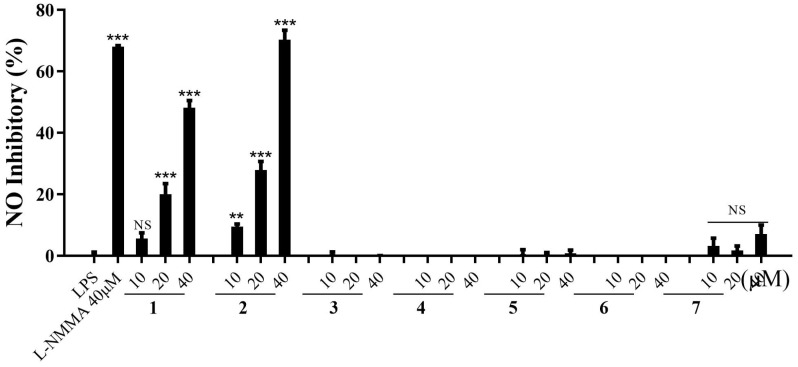
Inhibitory effects of compounds **1**–**7** on NO production in LPS-induced RAW 264.7 cells. Experiments were performed in triplicate, and the data are presented as the mean ± SD. ** *p* < 0.01, *** *p* < 0.001 vs. LPS group. NS, not significant by *T*-test.

**Table 1 molecules-28-00886-t001:** ^13^C NMR data of the aglycones of compounds **1**–**7** in pyridine-*d_5_* (125 MHz, δ in ppm).

Position	1 ^a^	2 ^a^	3 ^a^	4	5	6	7
1	37.4	37.6	38.5	37.8	38.1	37.9	37.8
2	29.1	29.1	28.9	29.8	29.6	29.8	29.7
3	76.4	76.4	76.6	77.8	77.9	76.0	75.9
4	34.7	34.8	34.5	35.2	35.1	35.2	35.2
5	44.0	44.0	44.5	44.0	43.9	43.9	43.9
6	26.7	26.7	27.0	27.2	27.1	27.2	27.3
7	31.8	31.9	32.3	25.2	25.2	25.1	25.3
8	66.9	66.9	66.0	66.8	66.8	66.7	66.9
9	51.3	51.3	54.3	51.9	51.8	51.7	51.8
10	39.1	39.2	39.3	39.4	39.4	39.4	39.5
11	68.8	69.5	68.6	69.1	68.8	69.0	69.9
12	75.2	75.5	74.2	75.2	75.2	74.9	75.5
13	46.1	46.2	47.3	44.0	46.1	46.1	46.5
14	71.5	71.6	71.6	71.7	71.6	71.7	71.9
15	26.8	26.8	27.7	32.2	32.1	32.1	32.1
16	25.1	25.1	25.4	27.1	27.1	27.1	27.2
17	59.8	59.9	60.3	59.7	60.0	59.9	59.9
18	16.6	16.7	17.5	16.9	17.0	16.9	16.9
19	12.7	12.8	12.9	13.2	13.2	13.1	13.2
20	211.1	211.1	212.6	210.3	210.0	210.1	210.4
21	30.3	30.2	32.7	30.3	29.9	30.0	30.2
11-*O*	Tig	Bz	HPA	Tig	mBu	HPA	Bz
1	167.4	166.2	172.0	167.2	175.5	171.5	166.5
2	128.5	130.0	41.1	129.2	41.5	41.4	130.6
3	138.1	129.5	125.3	138.1	26.6	124.6	129.9
4	11.6	128.1	130.7	12.2	11.9	131.3	128.7
5	14.2	132.9	115.6	14.3	15.7	116.4	133.5
6		128.1	155.2			158.2	128.7
7		129.5	115.6			116.4	129.9
8			130.7			131.3	
12-*O*	Bz	Bz		Ac	Ac	Ac	Bz
1	166.1	166.1		170.7	170.8	170.8	166.3
2	129.4	129.0		20.5	20.9	20.3	130.6
3	129.7	129.5					129.8
4	128.3	128.1					128.7
5	133.1	132.8					133.3
6	128.3	128.1					128.7
7	129.7	129.5					129.8

^a^ Measured in CDCl3.

**Table 2 molecules-28-00886-t002:** ^13^C NMR data of the sugar moieties of compounds **1**–**7** in pyridine-*d_5_* (125 MHz, δ in ppm).

Position	1 ^a^	2 ^a^	3 ^a^	4	5	6	7
	Ole	Ole	Ole	Glc-1	Glc-1	Ole	Ole
1	96.9	97.0	97.0	101.2	101.1	97.5	97.4
2	36.1	36.1	36.1	71.4	71.5	37.9	37.8
3	78.8	78.8	78.8	78.0	78.0	79.7	79.6
4	79.1	79.1	79.1	84.7	84.7	83.4	83.3
5	71.4	71.4	71.4	78.0	77.9	71.9	71.9
6	18.6	18.6	18.6	62.6	62.6	19.1	19.0
3-OMe	55.6	55.6	55.7			57.3	57.3
	Allo	Allo	Allo	Glc-2	Glc-2	Allo	Allo
1	99.1	99.1	99.2	106.7	106.7	102.0	101.9
2	71.8	71.8	71.8	77.1	77.1	72.7	72.7
3	81.0	81.0	81.0	78.3	78.3	83.2	83.2
4	72.8	72.8	72.8	71.4	71.5	83.4	83.3
5	71.3	71.3	71.3	78.9	78.9	69.6	69.5
6	17.9	17.9	17.9	62.6	62.6	18.4	18.3
3-OMe	62.0	61.9	62.0			61.7	61.7
						Glc	Glc
1						106.6	106.6
2						75.5	75.5
3						78.4	78.4
4						72.0	72.0
5						78.5	78.4
6						63.0	63.0

^a^ Measured in CDCl_3_.

**Table 3 molecules-28-00886-t003:** ^1^H NMR data of the sugar moieties of compounds **1**–**7** in pyridine-*d_5_* (500 MHz, δ in ppm, *J* in Hz).

Position	1 ^a^	2 ^a^	3 ^a^	4	5	6	7
	Ole	Ole	Ole	Glc-1	Glc-1	Ole	Ole
1	4.58 dd	4.58 dd	4.58 dd	5.01 d	5.08 d	4.78 d	4.73 d
	(9.8, 1.8)	(9.8, 1.9)	(9.7, 1.7)	(7.6)	(7.7)	(8.9)	(9.4)
2	1.47 m	1.45 m	1.49 m	4.31 m	4.31 m	1.26 m	1.33 m
	2.30 m	2.30 m	2.30 m			2.41 m	2.36 m
3	3.39 m	3.38 m	3.38 m	4.25 m	4.23 m	3.61 m	3.57 m
4	3.33 m	3.31 m	3.34 m	4.09 m	4.13 m	3.59 m	3.52 m
5	3.33 m	3.31 m	3.34 m	3.92 m	3.92 m	3.64 m	3.52 m
6	1.36 d	1.32 d	1.37 d	4.33 m	4.34 m	1.67 d	1.57 d
	(5.5)	(5.5)	(5.5)	4.50 m	4.52 m	(5.8)	(5.0)
3-OMe	3.37 s	3.35 s	3.37 s			3.50 s	3.47 s
	Allo	Allo	Allo	Glc-2	Glc-2	Allo	Allo
1	4.79 d	4.77 d	4.79 d	5.22 d	5.25 d	5.27 d	5.24 d
	(8.1)	(8.3)	(8.3)	(7.8)	(7.8)	(8.1)	(8.1)
2	3.47 m	3.46 m	3.47 m	4.11 m	4.12 m	3.84 m	3.79 m
3	3.79 t	3.78 t	3.78 t	4.35 m	4.37 m	4.47 t	4.45 m
	(3.0)	(3.0)	(3.0)			(2.5)	
4	3.17 m	3.17 m	3.17 m	4.21 m	4.21 m	3.73 dd	3.72 dd
						(9.6, 2.5)	(9.4, 2.0)
5	3.55 m	3.54 m	3.55 m	3.93 m	3.94 m	4.27 m	4.23 m
6	1.25 d	1.24 d	1.25 d	4.33 m	4.34 m	1.63 d	1.61 d
	(6.0)	(6.4)	(6.1)	4.45 m	4.43 m	(6.2)	(6.2)
3-OMe	3.66 s	3.65 s	3.65 s			3.81 s	3.80 s
						Glc	Glc
1						4.97 d	4.95 d
						(7.8)	(7.7)
2						4.02 m	4.00 m
3						4.26 m	4.22 m
4						4.22 m	4.19 m
5						4.01 m	3.97 m
6						4.36 dd	4.36 dd
						(11.6, 5,4)	(11.6, 5,1)
						4.54 dd	4.53 d
						(11.6, 2.3)	(11.6)

^a^ Measured in CDCl_3_.

**Table 4 molecules-28-00886-t004:** Inhibitory effects of compounds **1**–**7** on NO production in LPS-induced RAW 264.7 cells.

Compound	Concentration (μM)	NO Inhibition Rate (%)
**1**	40	48.19 ± 4.14
**2**	40	70.33 ± 5.39
**3**	40	−4.09 ± 7.28
**4**	40	−0.86 ± 1.59
**5**	40	0.80 ± 1.91
**6**	40	−5.57 ± 1.15
**7**	40	7.13 ± 5.00
L-NMMA ^a^	40	68.03 ± 0.72

^a^ Positive control.

## Data Availability

All the data in this research were presented in the manuscript and Appendix A.

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
