# Peer review of "Polyoxypregnane Glycosides from Root of Marsdenia tenacissima and Inhibited Nitric Oxide Levels in LPS Stimulated RAW 264.7 Cells"

_molecules, 2023, doi:10.3390/molecules28020886_

Round 1

Reviewer 1 Report

In this study, the authors attempted to characterize and determine the anti-inflammatory effect of ‘Polyoxypregnane glycosides steroids’ from Marsdenia tenacissima root. Although the authors characterized new polyoxypregnane glycosides through their comprehensive spectroscopic data, they failed majorly in determining their molecular mechanisms behind the anti-inflammatory effect. The data presented in figure-3 is not sufficient and requires revision. Overall, the study possess significance and the authors need to perform few more experiments to prove their hypothesis.

The following concerns need to be addressed point-by-point:

  1. The abstract and introduction must be re-written as the information provided is too short.
  1. A graphical abstract and overall scheme will help the readers.
  2. Why the authors used only ‘ethanol’ for the extraction of glycosides? Why they not attempted using water/methanol?
  3. Did the authors perform any ‘Glycosides identification test’ before characterization?
  4. What is the passage number of RAW264.7 macrophages used in this study?
  5. How the authors fixed the dose of LPS as 1μg/mL?
  6. In methodology section 3.5; line-379, the authors mentioned ‘The cell viability was determined by MTS assay..’. However, there no result or discussion for the same is provided?
  7. The claim of ‘Anti-inflammatory effect’ by measuring only ‘NO levels’ is not acceptable. Hence, the authors must determine the effect of compounds on other cytokines such as TNF-alpha, IL-6, IL-10.
  8. In line-224, the authors mentioned ‘No inhibitory was shown in Figure 4’. This must be revised to ‘NO inhibitory was shown in Figure 3’.
  9. In Figure 3, the compounds 1 & 2 showed dose dependent NO inhibitory activity and other compounds did not showed activity. The authors must discuss in detail this with relevant citations.
  1. In Figure 3, there is no mention about the statistical analysis?  How many times the assay was performed?

Reviewer 2 Report

Dear Editor,

Enclosed please find the comments on the following manuscript:

Journal: molecules

Manuscript ID: molecules-2115637

Title:Polyoxypregnane Glycosides Steroids from Root of Marsdenia tenacissima and Their Anti-Inflammatory Activity

The manuscript “Polyoxypregnane Glycosides Steroids from Root of Marsdenia tenacissima and Their Anti-Inflammatory Activityfirst report six new polyoxypregnane glycosides isolated from the roots of Marsdenia tenacissimahe by Zhi Na et. al, is a good research article in a current and interesting topic and could be of interest for traditional herbal medicine and phytochemistry scientists. In my opinion the manuscript is well written, the methodology and results clearly described. The results reported are of contribution for pharmaceutical science. In my opinion, there are minor concerns that impair the publication of this review in the present form.

Minor corrections:

1.The first word of keywords in research articles should be capitalized

Page 1, line14: Anti-inflammatory;

2. The scitific name of any species of plants should almost always be italicized.

Page 2, line 60 and 70: M. tenacissima.

The text editing of article should be consistently. Authors are advised to re-edit paragraphs of the following articles:

Page 3, line 77: The cross-peaks between δH 4.79 (H-1 of Allo) and δC 79.1 (C-4 of Ole); between δH 4.58……

Page 4, line 84-85: dated as 3-O-6-deoxy-3-O-methyl-β-D-allopyranosyl (1→4)-β-D-oleandropyranosyl-11α-O-tigloyl-12β-O-benzoyl-tenacigenin B and named……

Page 4, line 105-106: this Bz group were established. Therefore, 2 was defined as 3-O-6-deoxy-3-O-methyl-β-D-allopyanosyl(1→4)-β-D-oleandropyranosyl-11α,12β-di-O-……

Page 5, line 127-128: belonging to the HPA group. Therefore, 3 was defined as 3-O-6-deoxy-3-O-methyl-β-D-allopyanosyl(1→4)-β-D-oleandropyranosyl-11α-O-(4-hydr…..

Page 6, line 157-158: Glc1-H-1/aglycone-C-3. As a result, structure of 4 was finally elucidated as 3-O-β-D-glucopyranosyl-(1→4)-β-D-glucopyranosyl-11α-O-tigloyl-12β-O-acetyl-……

Page 6, line 173-174: structure of 5 was established as 3-O-β-D-glucopyranosyl-(1→4)-β-D-glucopyranosyl-11α-O-2-methylbutyryl-12β-O-acet……

Page 6, line 184-185: of the acetyl group. Thus, the aglycone of 6 was determined to be 11α-O-(4-hydroxyphenyl) acetyl-12β-O-acetyl-tenacigenin B…….

Page 7, line 199-200: The sequence of the sugar units was deduced as β-D-glucopyranosyl-(1→4)-6-deoxy-3-O-methyl-β-D-allopyranosyl-(1→4)-β-D-oleandro……

Page 7, line 207-208: δC 76.0 (aglycone-C-3). Consequently, the structure of 6 was elucidated as 207 3-O-β-D-glucopyranosyl-(1→4)-6-deoxy-3-O-methyl-β-D-allopyranosyl-(1→4)-β-D-olea……

Page 7, line 216-217: were reported. The structure of 7 was elucidated as 3-O-β-D-glucopyranosyl-(1→4)-6-deoxy-3-O-methyl-β-D-allopyranosyl-(1→4)-β-D-olea……

3. New compounds separation weight is missing a comma.

Page 8, line 265, 268 and 273: 6 (9 mg, tR = 19.6 min), (8 mg, tR = 9.6 min), 1 (9 mg, tR = 16.6 min) and 2 (10 mg, tR = 20.7 min) ……

4. The Monocyte/macrophage cell lines should have an extra space before the number.

Page 10, line 370 and 373: The RAW 264.7……

Reviewer 3 Report

General comment

The manuscript titled “Polyoxypregnane Glycosides Steroids from Root of Marsdenia tenacissima and Their Anti-Inflammatory Activity” described the isolation of seven polyoxypregnane glycosides of which six are new. The isolated compounds were tested for their inhibitory effect on LPS-induced NO production in RAW 264.7 cells. The data is sufficiently and comprehensively represented, however, there are many typos that should be corrected before publication.

Comments to authors

Page 1; line 16; remove the extra semicolon “;” at the end of the keywords statement.

Page 2; Lines 60 & 70 and page 7; line 213, please use italic font for the Latin binomial plant name.

Line 25; please change few study....to...few studies.

The introduction is short and highly similar to that of Pang et al., 2015. I suggest adding a section to the introduction on NO production and its contribution to inflammation and several pathogeneses with citing supporting references.

Line 41; the coupling constant symbol “J” should be in italic font all over the document (i.e., this point forward).

Line 45; 1.46(s).... add a space.

In figure 2; use the same red color for the curved key arrow that refers to the HMBC correlations (i.e., red colored arrow).

Line 58, 84-85, 75, 79, 106, 128, 143, 154, 174, 185, 195, 200, 208, 209, 217, 218 use italic font for the “O” atom in all glycosidic linkages.

Lines 72, 108; pyridine-d5... should be in italic font for “d5”.

Line 74; The authors use oleandropyranosyl (Ole) in text, but in Table 2 there is typing mistake as it was written as “Olc”, please correct.

The authors sometimes use hyphen writing NMR methods, e.g., in Line 39; 1H-NMR, line 160; 13C-NMR & sometimes without hyphens as in lines 91, 112, 165, 186; 1H NMR, line 81, 91; 13C NMR,...etc. Please revise and unify the style.

Line 108; 13C NMR data of the sugar moietie...correct to “moieties”.

Line 111; in accord....correct to in accordance.

Line 111; the authors mentioned that compound 3 showed a quasi-molecular ion peak at m/z 883.3889 ([M+Na]+)...remove round brackets, in accord with the molecular formula C43H62O14 (calcd for C43H62NaO14, 825.4032) but in Figure S15 the observed quasi-molecular ion appeared at m/z 825.4040, please revise and correct the observed value.

Line 145; (d, 1H, J = 7.6) and 5.22 (d, 1H, J = 7.8)....add Hz as the coupling constant unit.

Lines 160, 161; the authors stated that “The 13C-NMR data due to the sugar moieties of 5 agreed well with those of 4, which provided the evidence that 4 had the same sugar moiety as 1”. There is a mistake here since the sugar moieties in 4 and 1 are different! ..... it should be corrected to ...the same as 5.

Line 188; HMQC spectrum.... should be HSQC (see Figure S40).

Line 195; 6-deoxy-3-O-methyl-allose,[23]....remove the comma between allose and the reference number.

Line 224; No inhibitory was shown.... should be NO inhibition or NO inhibitory effect, with correcting nitric oxide formula to “NO” and adding the word “effect” as well.

Similarly, in line 227; Figure 3. No inhibitory of Compounds 17. It should be changed to “NO inhibitory effects of compounds 17”. Also, the name of the biological assay should be mentioned (i.e., using lipopolysaccharide-induced NO production in RAW 264.7) both in the figure caption and table title.

In lines 261-262; the authors mentioned that Fr. B.2 (2.8 g) was chromatographed over a Sephadex LH-20 column, eluting with MeOH to give four fractions. What was the solvent system used? Did you use methanol only as the solvent for purification on Sephadex LH-20?

Lines 255-273; section 3.3. “Extraction and isolation”, all isolated compound numbers (codes) should be written in bold font.

Line 276; λmax....max should be subscript font.

Page 9; lines 275-313; in section “3.4. Compound Characterization Data”, all compounds’ numbers/codes (i.e., compounds 1-5) should be written in bold font.

Line 305; 21-CH3 .... 21-CH3 (i.e., subscript font).

Line 392; preciously......previously.

Line 393; for anti-inflammatory........ for anti-inflammatory activity or effect.

Round 2

Reviewer 1 Report

The author’s response for the queries raised was checked point by point and summarized below:

  1. For query-1, the authors response is satisfactory.
  2. For query-2, the authors response is satisfactory.
  3. For query-3, the authors response is reasonable.
  4. For query-4, the authors response is acceptable and must include the Liebermann Burchard result as a supplementary figure.  
  5. For query-5, the authors response is acceptable and must include the passage number detail in line number 409.
  6. For query-6, the authors response is acceptable and must include the reference in line number 412.
  7. For query-7, the authors response is acceptable.
  8. For query-8, since the authors did not observe satisfactory results for the inflammatory cytokines, the title of the manuscript must be revised to ‘Polyoxypregnane Glycosides from Root of Marsdenia tenacissima inhibited Nitric oxide levels in LPS stimulated RAW 264.7 cells’.
  9. For query-9, the authors response is acceptable.
  10. For query-10, the authors response is acceptable and must include this as discussion in section 2.2.
  11. For query-11, the authors response is satisfactory.  

I suggest the authors to revise the title of the manuscript and include the citations as mentioned above. Overall, the authors addressed satisfactory response to the queries raised by the reviewer and therefore I recommend the manuscript for publication. 

Author Response

The author’s response for the queries raised was checked point by point and summarized below:

For query-1, the authors response is satisfactory.

For query-2, the authors response is satisfactory.

For query-3, the authors response is reasonable.

For query-4, the authors response is acceptable and must include the Liebermann Burchard result as a supplementary figure. 

For query-5, the authors response is acceptable and must include the passage number detail in line number 409.

Response: The passage number of the RAW264.7 cells had been inserted in line 409.

For query-6, the authors response is acceptable and must include the reference in line number 412.

Response: The reference had been inserted in line 412.

For query-7, the authors response is acceptable.

For query-8, since the authors did not observe satisfactory results for the inflammatory cytokines, the title of the manuscript must be revised to ‘Polyoxypregnane Glycosides from Root of Marsdenia tenacissima inhibited Nitric oxide levels in LPS stimulated RAW 264.7 cells’.

Response: The title had been revised to “Polyoxypregnane Glycosides from Root of Marsdenia tenacissima and Inhibited Nitric Oxide Levels in LPS Stimulated RAW 264.7 cells”.

For query-9, the authors response is acceptable.

For query-10, the authors response is acceptable and must include this as discussion in section 2.2.

Response: The discussion had been added in section 2.2.

For query-11, the authors response is satisfactory. 

I suggest the authors to revise the title of the manuscript and include the citations as mentioned above. Overall, the authors addressed satisfactory response to the queries raised by the reviewer and therefore I recommend the manuscript for publication.

Reviewer 3 Report

The authors have responded to most of the raised comments, however, minor missed comments were found as follows:

Line 185; of 11α-O-tigloyl-12β-O-acetyl-tenacigenin B.. use italic font for the “O” atom in all glycosidic linkages.

Line 293; The authors did not respond to the following comment:

 Fr. B.2 (2.8 g) was chromatographed over a Sephadex LH-20 column, eluting with MeOH to give four fractions. What was the method used i.e., lipophilic or hydrophilic? Isocratic or gradient? i.e., did the author use chloroform-MeOH or water-MeOH to reach the 100% MeOH composition? ....this will allow the reproducibility of the method by future studies.

Author Response

The authors have responded to most of the raised comments, however, minor missed comments were found as follows:

Line 185; of 11α-O-tigloyl-12β-O-acetyl-tenacigenin B. use italic font for the “O” atom in all glycosidic linkages.

Response: This error had already been pointed out by the reviewer in round 1. Due to the authors’ carelessness, we miss it in line 185. The “O” atom in all glycosidic linkages had been written in italic font.

Line 293; The authors did not respond to the following comment:

 Fr. B.2 (2.8 g) was chromatographed over a Sephadex LH-20 column, eluting with MeOH to give four fractions. What was the method used i.e., lipophilic or hydrophilic? Isocratic or gradient? i.e., did the author use chloroform-MeOH or water-MeOH to reach the 100% MeOH composition? ....this will allow the reproducibility of the method by future studies.

Response: I'm sorry for not saying it exactly in round 1. “eluting with MeOH” means pure methanol is used as eluent from the beginning to the end, i.e. Isocratic elution.

In our lab, when separating compounds with a small polarity using Sephadex LH-20 column, we usually used CH2Cl2–MeOH (1:1) as eluent and isocratic elution. When separating compounds with a large polarity, we usually used pure MeOH as eluent and isocratic elution.